# Meta-DMoE: Adapting to Domain Shift by Meta-Distillation from Mixture-of-Experts

**Tao Zhong**[*1], **Zhixiang Chi**[*2], **Li Gu**[* 2], **Yang Wang**[2,3], **Yuanhao Yu**[2], **Jin Tang**[2]

[1]University of Toronto,   [2]Huawei Noah's Ark Lab,   [3]Concordia University
tao.zhong@mail.utoronto.ca   yang.wang@concordia.ca
{zhixiang.chi, li.gu, yuanhao.yu, tangjin}@huawei.com

## Abstract

In this paper, we tackle the problem of domain shift. Most existing methods perform training on multiple source domains using a single model, and the same trained model is used on all unseen target domains. Such solutions are sub-optimal as each target domain exhibits its own specialty, which is not adapted. Furthermore, expecting single-model training to learn extensive knowledge from multiple source domains is counterintuitive. The model is more biased toward learning only domain-invariant features and may result in negative knowledge transfer. In this work, we propose a novel framework for unsupervised test-time adaptation, which is formulated as a knowledge distillation process to address domain shift. Specifically, we incorporate Mixture-of-Experts (MoE) as teachers, where each expert is separately trained on different source domains to maximize their specialty. Given a test-time target domain, a small set of unlabeled data is sampled to query the knowledge from MoE. As the source domains are correlated to the target domains, a transformer-based aggregator then combines the domain knowledge by examining the interconnection among them. The output is treated as a supervision signal to adapt a student prediction network toward the target domain. We further employ meta-learning to enforce the aggregator to distill positive knowledge and the student network to achieve fast adaptation. Extensive experiments demonstrate that the proposed method outperforms the state-of-the-art and validates the effectiveness of each proposed component. Our code is available at https://github.com/n3il666/Meta-DMoE.

## 1   Introduction

The emergence of deep models has achieved superior performance [32, 40, 47]. Such unprecedented success is built on the strong assumption that the training and testing data are highly correlated (i.e., they are both sampled from the same data distribution). However, the assumption typically does not hold in real-world settings as the training data is infeasible to cover all the ever-changing deployment environments [39]. Reducing such distribution correlation is known as distribution shift, which significantly hampers the performance of deep models. Humans are more robust against the distribution shift, but artificial learning-based systems suffer more from performance degradation.

One line of research aims to mitigate the distribution shift by exploiting some unlabeled data from a target domain, which is known as unsupervised domain adaptation (UDA) [24, 51, 26]. The unlabeled data is an estimation of the target distribution [86]. Therefore, UDA normally adapts to the target domain by transferring the source knowledge via a common feature space with less effect from domain discrepancy [79, 50]. However, UDA is less applicable for real-world scenarios

---

[*]equal contribution

36th Conference on Neural Information Processing Systems (NeurIPS 2022).

as repetitive large-scale training is required for every target domain. In addition, collecting data samples from a target domain in advance might be unavailable as the target distribution could be unknown during training. Domain generalization (DG) [54, 28, 6] is an alternative line of research but more challenging as it assumes the prior knowledge of the target domains is unknown. DG methods leverage multiple source domains for training and directly use the trained model on all unseen domains. As the domain-specific information for the target domains is not adapted, a generic model is sub-optimal [68, 17].

Test-time adaptation with DG allows the model to exploit the unlabeled data during testing to overcome the limitation of using a flawed generic model for all unseen target domains. In ARM [86], meta-learning [25] is utilized for training the model as an initialization such that it can be adapted using the unlabeled data from the unseen target domain before making the final inference. However, we observed that ARM only trains a single model, which is counterintuitive for the multi-source domain setting. There is a certain amount of correlations among the source domains while each of them also exhibits its own specific knowledge. When the number of source domains rises, data complexity dramatically increases, which impedes thorough exploration of the dataset. Furthermore, real-world domains are not always balanced in data scales [39]. Therefore, the single-model training is more biased toward the domain-invariant features and dominant domains instead of the domain-specific features [12].

In this work, we propose to formulate the test-time adaptation as the process of knowledge distillation [34] from multiple source domains. Concretely, we propose to incorporate the concept of Mixture-of-Experts (MoE), which is a natural fit for the multi-source domain settings. The MoE models are treated as a teacher and separately trained on the corresponding domain to maximize their domain specialty. Given a new target domain, a few unlabeled data are collected to query the features from expert models. A transformer-based knowledge aggregator is proposed to examine the interconnection among queried knowledge and aggregate the correlated information toward the target domain. The output is then treated as a supervision signal to update a student prediction network to adapt to the target domain. The adapted student is then used for subsequent inference. We employ bi-level optimization as meta-learning to train the aggregator at the meta-level to improve generalization. The student network is also meta-trained to achieve fast adaptation via a few samples. Furthermore, we simulate the test-time out-of-distribution scenarios during training to align the training objective with the evaluation protocol.

The proposed method also provides additional advantages over ARM: 1) Our method provides a larger model capability to improve the generalization power; 2) Despite the higher computational cost, only the adapted student network is kept for inference, while the MoE models are discarded after adaptation. Therefore, our method is more flexible in designing the architectures for the teacher or student models. (e.g., designing compact models for the power-constrained environment); 3) Our method does not need to access the raw data of source domains but only needs their trained models. So, we can take advantage of private domains in a real-world setting where their data is inaccessible.

We name our method as **Meta-D**istillation of **MoE** (Meta-DMoE). Our contributions are as follows:

- We propose a novel unsupervised test-time adaptation framework that is tailored for multiple sources domain settings. Our framework employs the concept of MoE to allow each expert model to explore each source domain thoroughly. We formulate the adaptation process as knowledge distillation via aggregating the positive knowledge retrieved from MoE.

- The alignment between training and evaluation objectives via meta-learning improves the adaptation, hence the test-time generalization.

- We conduct extensive experiments to show the superiority of the proposed method among the state-of-the-arts and validate the effectiveness of each component of Meta-DMoE.

- We validate that our method is more flexible in real-world settings where computational power and data privacy are the concerns.

## 2 Related work

**Domain shift.** Unsupervised Domain Adaptation (UDA) has been popular to address domain shift by transferring the knowledge from the labeled source domain to the unlabeled target domain [48, 41, 81]. It is achieved by learning domain-invariant features via minimizing statistical discrepancy across

domains [5, 58, 70]. Adversarial learning is also applied to develop indistinguishable feature space [26, 51, 57]. The first limitation of UDA is the assumption of the co-existence of source and target data, which is inapplicable when the target domain is unknown in advance. Furthermore, most of the algorithms focus on unrealistic single-source-single-target adaptation as source data normally come from multiple domains. Splitting the source data into various distinct domains and exploring the unique characteristics of each domain and the dependencies among them strengthen the robustness [88, 76, 78]. Domain generalization (DG) is another line of research to alleviate the domain shift. DG aims to train a model on multiple source domains without accessing any prior information of the target domain and expects it to perform well on unseen target domains. [28, 45, 53] aim to learn the domain-invariant feature representation. [63, 75] exploit data augmentation strategies in data or feature space. A concurrent work proposed bidirectional learning to mitigate domain shift [14]. However, deploying the generic model to all unseen target domains fails to explore domain specialty and yields sub-optimal solutions. In contrast, our method further exploits the unlabeled target data and updates the trained model to each specific unseen target domain at test time.

**Test-time adaptation (TTA).** TTA constructs supervision signals from unlabeled data to update the generic model before inference. Sun *et al.* [68] uses rotation prediction to update the model during inference. Chi *et al.* [17] and Li *et al.* [46] reconstruct the input images to achieve internal-learning to restore the blurry images and estimate the human pose. ARM [86] incorporates test-time adaptation with DG, which meta-learns a model that is capable of adapting to unseen target domains before making an inference. Instead of adapting to every data sample, our method only updates once for each target domain using a fixed number of examples.

**Meta-learning.** The existing meta-learning methods can be categorised as model-based [62, 59, 8], metric-based [65, 30], and optimization-based [25]. Meta-learning aims to learn the learning process by episodic learning, which is based on bi-level optimization ([13] provides a comprehensive survey). One of the advantages of bi-level optimization is to improve the training with conflicting learning objectives. Utilizing such a paradigm, [16, 85] successfully reduce the forgetting issue and improve adaptation for continual learning [49]. In our method, we incorporate meta-learning with knowledge distillation by jointly learning a student model initialization and a knowledge aggregator for fast adaptation.

**Mixture-of-experts.** The goal of MoE [37] is to decompose the whole training set into many subsets, which are independently learned by different models. It has been successfully applied in image recognition models to improve the accuracy [1]. MoE is also popular in scaling up the architectures. As each expert is independently trained, sparse selection methods are developed to select a subset of the MoE during inference to increase the network capacity [42, 23, 29]. In contrast, our method utilizes all the experts to extract and combine the knowledge for positive knowledge transfer.

## 3 Preliminaries

In this section, we describe the problem setting and discuss the adaptive model. We mainly follow the test-time unsupervised adaptation as in [86]. Specifically, we define a set of $N$ source domains $\mathcal{D}_{\mathcal{S}} = \{\mathcal{D}_{\mathcal{S}}^i\}_{i=1}^N$ and $M$ target domains $\mathcal{D}_{\mathcal{T}} = \{\mathcal{D}_{\mathcal{T}}^j\}_{j=1}^M$. The exact definition of a domain varies and depends on the applications or data collection methods. It could be a specific dataset, user, or location. Let $x \in \mathcal{X}$ and $y \in \mathcal{Y}$ denote the input and the corresponding label, respectively. Each of the source domains contains data in the form of input-output pairs: $\mathcal{D}_{\mathcal{S}}^i = \{(x_{\mathcal{S}}^z, y_{\mathcal{S}}^z)\}_{z=1}^{Z_i}$. In contrast, each of the target domains contains only unlabeled data: $\mathcal{D}_{\mathcal{T}}^j = \{(x_{\mathcal{T}}^k)\}_{k=1}^{K_j}$. For well-designed datasets (e.g. [33, 20]), all the source or target domains have the same number of data samples. Such condition is not ubiquitous for real-world scenarios (i.e. $Z_{i_1} \neq Z_{i_2}$ if $i_1 \neq i_2$ and $K_{j_1} \neq K_{j_2}$ if $j_1 \neq j_2$) where data imbalance always exists [39]. It further challenges the generalization with a broader range of real-world distribution shifts instead of finite synthetic ones. Generic domain shift tasks focus on the out-of-distribution setting where the source and target domains are non-overlapping (i.e. $\mathcal{D}_{\mathcal{S}} \cap \mathcal{D}_{\mathcal{T}} = \varnothing$), but the label spaces of both domains are the same (i.e. $\mathcal{Y}_{\mathcal{S}} = \mathcal{Y}_{\mathcal{T}}$).

Conventional DG methods perform training on $\mathcal{D}_{\mathcal{S}}$ and make a minimal assumption on the testing scenarios [67, 3, 35]. Therefore, the same generic model is directly applied to all target domains $\mathcal{D}_{\mathcal{T}}$, which leads to sub-optimal solutions [68]. In fact, for each $\mathcal{D}_{\mathcal{T}}^j$, some unlabeled data are readily available which provides certain prior knowledge for that target distribution. Adaptive Risk Minimization (ARM) [86] assumes that a batch of unlabeled input data **x** approximate the

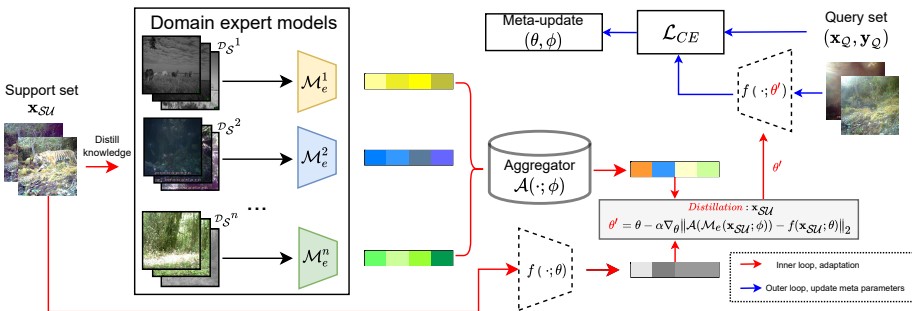

Figure 1: Overview of the training of Meta-DMoE. We first sample disjoint support set $\mathbf{x}_{\mathcal{SU}}$ and query set $(\mathbf{x}_{\mathcal{Q}}, \mathbf{y}_{\mathcal{Q}})$ from a training domain. $\mathbf{x}_{\mathcal{SU}}$ is sent to the expert models $\mathcal{M}$ to query their domain-specific knowledge. An aggregator $\mathcal{A}(\cdot; \phi)$ then combines the information and generates a supervision signal to update the $f(\cdot; \theta)$ via knowledge distillation. The updated $f(\cdot; \theta')$ is evaluated using the labeled query set to update the meta-parameters.

input distribution $p_x$ which provides useful information about $p_{y|x}$. Based on the assumption, an unsupervised test-time adaptation [59, 27] is proposed. The fundamental concept is to adapt the model to the specific domain using $\mathbf{x}$. Overall, ARM aims to minimize the following objective $\mathcal{L}(\cdot, \cdot)$ over all training domains:

$$\sum_{\mathcal{D}_{\mathcal{S}}^{j} \in \mathcal{D}_{\mathcal{S}}} \sum_{(\mathbf{x}, \mathbf{y}) \in \mathcal{D}_{\mathcal{S}}^{j}} \mathcal{L}(\mathbf{y}, f(\mathbf{x}; \theta')), \text{where } \theta' = h(\mathbf{x}, \theta; \phi). \tag{1}$$

$\mathbf{y}$ is the labels for $\mathbf{x}$. $f(\mathbf{x}; \theta)$ denotes the prediction model parameterized by $\theta$. $h(\cdot; \phi)$ is an adaptation function parameterized by $\phi$. It receives the original $\theta$ of $f$ and the unlabeled data $\mathbf{x}$ to adapt $\theta$ to $\theta'$.

The goal of ARM is to learn both $(\theta, \phi)$. To mimic the test-time adaptation (i.e., adapt before prediction), it follows the episodic learning as in meta-learning [25]. Specifically, each episode processes a domain by performing unsupervised adaptation using $\mathbf{x}$ and $h(\cdot; \phi)$ in the inner loop to obtain $f(\cdot; \theta')$. The outer loop evaluates the adapted $f(\cdot; \theta')$ using the true label to perform a meta-update. ARM is a general framework that can be incorporated with existing meta-learning approaches with different forms of adaptation module $h(\cdot; \cdot)$ [25, 27].

However, several shortcomings are observed with respect to the generalization. The episodic learning processes one domain at a time, which has clear boundaries among the domains. The overall setting is equivalent to the multi-source domain setting, which is proven to be more effective than learning from a single domain [53, 87] as most of the domains are correlated to each other [2]. However, it is counterintuitive to learn all the domain knowledge in one single model as each domain has specialized semantics or low-level features [64]. Therefore, the single-model method in ARM is sub-optimal due to: 1) some domains may contain competitive information, which leads to negative knowledge transfer [66]. It may tend to learn the ambiguous feature representations instead of capturing all the domain-specific information [80]; 2) not all the domains are equally important [76], and the learning might be biased as data in different domains are imbalanced in real-world applications [39].

## 4 Proposed approach

In this section, we explicitly formulate the test-time adaptation as a knowledge transfer process to distill the knowledge from MoE. The proposed method is learned via meta-learning to mimic the test-time out-of-distribution scenarios and ensure positive knowledge transfer.

### 4.1 Meta-distillation from mixture-of-experts

**Overview.** Fig. 1 shows the method overview. We wish to explicitly transfer useful knowledge from various source domains to achieve generalization on unseen target domains. Concretely, we define MoE as $\mathcal{M} = \{\mathcal{M}^i\}_{i=1}^{N}$ to represent the domain-specific models. Each $\mathcal{M}^i$ is separately trained using standard supervised learning on the source domain $\mathcal{D}_{\mathcal{S}}^{i}$ to learn its discriminative features.

We propose the test-time adaptation as the unsupervised knowledge distillation [34] to learn the knowledge from MoE. Therefore, we treat $\mathcal{M}$ as the teacher and aim to distill its knowledge to a student prediction network $f(\cdot; \theta)$ to achieve adaptation. To do so, we sample a batch of unlabeled $\mathbf{x}$ from a target domain, and pass it to $\mathcal{M}$ to query their domain-specific knowledge $\{\mathcal{M}^i(\mathbf{x})\}_{i=1}^N$. That knowledge is then forwarded to a knowledge aggregator $\mathcal{A}(\cdot; \phi)$. The aggregator is learned to capture the interconnection among domain knowledge and aggregate the information from MoE. The output of $\mathcal{A}(\cdot; \phi)$ is treated as the supervision signal to update $f(\mathbf{x}; \theta)$. Once the adapted $\theta'$ is obtained, $f(\cdot; \theta')$ is used to make predictions for the rest of the data in that domain. The overall framework follows the effective few-shot learning paradigm where $\mathbf{x}$ is treated as an unlabeled support set [74, 65, 25].

**Training Meta-DMoE.** Properly training $(\theta, \phi)$ is critical to improve the generalization on unseen domains. In our framework, $\mathcal{A}(\cdot, \phi)$ acts as a mechanism that explores and mixes the knowledge from multiple source domains. Conventional knowledge distillation process requires large numbers of data samples and learning iterations [34, 2]. The repetitive large-scale training is inapplicable in real-world applications. To mitigate the aforementioned challenges, we follow the meta-learning paradigm [25]. Such bi-level optimization enforces the $\mathcal{A}(\cdot, \phi)$ to learn beyond any specific knowledge [85] and allows the student prediction network

---

**Algorithm 1** Training for Meta-DMoE

**Require:** $\{\mathcal{D}_{\mathcal{S}}{}^i\}_{i=1}^N$: data of source domains; $\alpha, \beta$: learning rates; $B$: meta batch size
1: // Pretrain domain-specific MoE models
2: **for** $i=1,...,N$ **do**
3:     Train the domain-specific model $\mathcal{M}^i$ using $\mathcal{D}_{\mathcal{S}}{}^i$.
4: **end for**
5: // Meta-train aggregator $\mathcal{A}(\cdot, \phi)$ **and student model** $f(\cdot, \theta_e; \theta_c)$
6: **Initialize:** $\phi, \theta_e, \theta_c$
7: **while** *not converged* **do**
8:     Sample a batch of $B$ source domains $\{\mathcal{D}_{\mathcal{S}}{}^b\}^B$, reset batch loss $\mathcal{L}_{\mathcal{B}} = 0$
9:     **for** each $\mathcal{D}_{\mathcal{S}}{}^b$ **do**
10:         Sample support and query set: $(\mathbf{x}_{\mathcal{SU}}), (\mathbf{x}_{\mathcal{Q}}, \mathbf{y}_{\mathcal{Q}}) \sim \mathcal{D}_{\mathcal{S}}{}^b$
11:         $\mathcal{M}'_e(\mathbf{x}_{\mathcal{SU}}; \phi) = \{\mathcal{M}^i_e(\mathbf{x}_{\mathcal{SU}}; \phi)\}_{i=1}^N$, mask $\mathcal{M}^i_e(\mathbf{x}_{\mathcal{SU}}; \phi)$ with $\mathbf{0}$ if $b = i$
12:         Perform adaptation via knowledge distillation from MoE:
13:         $\theta'_e = \theta_e - \alpha\nabla_{\theta_e} \|\mathcal{A}(\mathcal{M}'_e(\mathbf{x}_{\mathcal{SU}}; \phi)) - f(\mathbf{x}_{\mathcal{SU}}; \theta_e)\|_2$
14:         Evaluate the adapted $\theta'$ using the query set and accumulate the loss:
15:         $\mathcal{L}_{\mathcal{B}} = \mathcal{L}_{\mathcal{B}} + \mathcal{L}_{CE}(\mathbf{y}_{\mathcal{Q}}, f(\mathbf{x}_{\mathcal{Q}}; \theta'_e, \theta_c))$
16:     **end for**
17:     Update $\phi, \theta_e, \theta_c$ for the current meta batch:
18:     $(\phi, \theta_e, \theta_c) \leftarrow (\phi, \theta_e, \theta_c) - \beta\nabla_{(\phi, \theta_e, \theta_c)}\mathcal{L}_{\mathcal{B}}$
19: **end while**

---

$f(\cdot; \theta)$ to achieve fast adaptation. Specifically, We first split the data samples in each source domain $\mathcal{D}_{\mathcal{S}}{}^i$ into disjoint support and query sets. The unlabeled support set $(\mathbf{x}_{\mathcal{SU}})$ is used to perform adaptation via knowledge distillation, while the labeled query set $(\mathbf{x}_{\mathcal{Q}}, \mathbf{y}_{\mathcal{Q}})$ is used to evaluate the adapted parameters to explicitly test the generalization on unseen data.

The student prediction network $f(\cdot; \theta)$ can be decoupled as a feature extractor $\theta_e$ and classifier $\theta_c$. Unsupervised knowledge distillation can be achieved via the softened output [34] or intermediate features [84] from $\mathcal{M}$. The former one allows the whole student network $\theta = (\theta_e, \theta_c)$ to be adaptive, while the latter one allows partial or complete $\theta_e$ to adapt to $\mathbf{x}$, depending on the features utilized. We follow [56] to only adapt $\theta_e$ in the inner loop while keeping the $\theta_c$ fixed. Thus, the adaptation process is achieved by distilling the knowledge via the aggregated features:

$$DIST(\mathbf{x}_{\mathcal{SU}}, \mathcal{M}_e, \phi, \theta_e) = \theta'_e = \theta_e - \alpha\nabla_{\theta_e} \|\mathcal{A}(\mathcal{M}_e(\mathbf{x}_{\mathcal{SU}}); \phi) - f(\mathbf{x}_{\mathcal{SU}}; \theta_e)\|_2, \qquad (2)$$

where $\alpha$ denotes the adaptation learning rate, $\mathcal{M}_e$ is the feature extractor of MoE models, which extracts the features before the classifier, and $\|\cdot\|_2$ measures the $L_2$ distance. The goal is to obtain an updated $\theta'_e$ such that the extracted features of $f(\mathbf{x}_{\mathcal{SU}}; \theta'_e)$ is closer to the aggregated features. The overall learning objective of Meta-DMoE is to minimize the following expected loss:

$$\arg\min_{\theta_e, \theta_c, \phi} \sum_{\mathcal{D}_{\mathcal{S}}{}^j \in \mathcal{D}_{\mathcal{S}}} \sum_{\substack{(\mathbf{x}_{\mathcal{SU}}) \in \mathcal{D}_{\mathcal{S}}{}^j \\ (\mathbf{x}_{\mathcal{Q}}, \mathbf{y}_{\mathcal{Q}}) \in \mathcal{D}_{\mathcal{S}}{}^j}} \mathcal{L}_{CE}(\mathbf{y}_{\mathcal{Q}}, f(\mathbf{x}_{\mathcal{Q}}; \theta'_e, \theta_c)), \text{where } \theta'_e = DIST(\mathbf{x}_{\mathcal{SU}}, \mathcal{M}_e, \phi, \theta_e),$$

$$(3)$$

where $\mathcal{L}_{CE}$ is the cross-entropy loss. Alg. 1 demonstrates our full training procedure. To smooth the meta gradient and stabilize the training, we process a batch of episodes before each meta-update.

Since the training domains overlap for the MoE and meta-training, we simulate the test-time out-of-distribution by excluding the corresponding expert model in each episode. To do so, we multiply the features by $\mathbf{0}$ to mask them out. $\mathcal{M}'_e$ in L11 of Alg. 1 denotes such operation. Therefore, the adaptation is enforced to use the knowledge that is aggregated from other domains.

## 4.2 Fully learned knowledge aggregator

Aggregating the knowledge from distinct domains requires capturing the relation among them to ensure the relevant knowledge transfer. Prior works design hand-engineered solutions to combine the knowledge or choose data samples that are closer to the target domain for knowledge transfer [2, 88]. A superior alternative is to replace the hand-designed pipelines with fully learned solutions [19, 9]. Thus we follow the same trend and allow the aggregator $\mathcal{A}(\cdot; \phi)$ to be fully meta-learned without heavy hand-engineering.

We observe that the self-attention mechanism is quite suitable where interaction among domain knowledge can be computed. Therefore, we use a transformer encoder as the aggregator [22, 73]. The encoder consists of multi-head self-attention and multi-layer perceptron blocks with layernorm [4] applied before each block and residual connection applied after each block. We refer the readers to the appendix for the detailed architecture and computation. We concatenate the output features from the MoE models as $Concat[\mathcal{M}_e^1(\mathbf{x}), \mathcal{M}_e^2(\mathbf{x}), ..., \mathcal{M}_e^N(\mathbf{x})] \in \mathbb{R}^{N \times d}$, where $d$ is the feature dimension. The aggregator $\mathcal{A}(\cdot; \phi)$ processes the input tensor to obtain the aggregated feature $\mathbf{F} \in \mathbb{R}^d$, which is used as a supervision signal for test-time adaptation.

## 4.3 More constrained real-world settings

In this section, we investigate two critical settings for real-world applications that have drawn less attention from the prior works: limitation on computational resources and data privacy.

**Constraint on computational cost.** In real-world deployment environments, the computational power might be highly constrained (e.g., smartphones). It requires fast inference and compact models. However, the reduction in learning capabilities greatly hinders the generalization as some methods utilize only a single model regardless of the data complexity. On the other hand, when the number of domain data scales up, methods relying on adaptation on every data sample [86] will experience inefficiency. In contrast, our method only needs to perform adaptation once for every unseen domain. Only the final $f(\cdot; \theta')$ is used for inference. To investigate the impact on generalization caused by reducing the model size, we experiment with some lightweight network architectures (only $f(\cdot; \theta)$ for us) such as MobileNet V2 [61].

**Data privacy.** Large-scale training data are normally collected from various venues. However, some venues may have privacy regulations enforced. Their data might not be accessible, but the models that are trained using private data are available. To simulate such an environment, we split the training source domains into two splits: private domains ($\mathcal{D}_{\mathcal{S}}^{pri}$) and public domains ($\mathcal{D}_{\mathcal{S}}^{pub}$). We use $\mathcal{D}_{\mathcal{S}}^{pri}$ to train MoE models and $\mathcal{D}_{\mathcal{S}}^{pub}$ for the subsequent meta-training. Since ARM and other methods only utilize the data as input, we train them on $\mathcal{D}_{\mathcal{S}}^{pub}$.

We conduct experiments to show the superiority of the proposed method in these more constrained real-world settings with computation and data privacy issues. For details on the settings, please refer to the supplementary materials.

# 5 Experiments

## 5.1 Datasets and implementation details

**Datasets and evaluation metrics.** In this work, we mainly evaluate our method on real-world domain shift scenarios. Drastic variation in deployment conditions normally exists in nature, such as a change in illumination, background, and time. It shows a huge domain gap between deployment environments and imposes challenges to the algorithm's robustness. Thus, we test our methods on the large-scale distribution shift benchmark WILDS [39], which reflects a diverse range of real-world distribution shifts. Following [86], we mainly perform experiments on five image testbeds, iWildCam [10], Camelyon17 [7],RxRx1 [69] and FMoW [18] and PovertyMap [83]. In each benchmark dataset, a domain represents a distribution of data that is similar in some way, such as images collected from the same camera trap or satellite images taken in the same location. We follow the same evaluation metrics as in [39] to compute several metrics: accuracy, Macro F1, worst-case (WC) accuracy, Pearson correlation (r), and its worst-case counterpart. We also evaluate our method on popular benchmarks DomainNet [58] and PACS [44] from DomainBed [31] by computing the accuracy.

Table 1: Comparison with the state-of-the-arts on the WILDS image testbeds and out-of-distribution setting. Metric means and standard deviations are reported across replicates. Our proposed method performs well across all problems and achieves the best results on 4 out of 5 datasets.

| | iWildCam | | Camelyon17 | RxRx1 | FMoW | | PovertyMap | |
|---|---|---|---|---|---|---|---|---|
| Method | Acc | Macro F1 | Acc | Acc | WC Acc | Avg Acc | WC Pearson r | Pearson r |
| ERM | 71.6 (2.5) | 31.0 (1.3) | 70.3 (6.4) | 29.9 (0.4) | 32.3 (1.25) | **53.0 (0.55)** | 0.45 (0.06) | 0.78 (0.04) |
| CORAL | 73.3 (4.3) | 32.8 (0.1) | 59.5 (7.7) | 28.4 (0.3) | 31.7 (1.24) | 50.5 (0.36) | 0.44 (0.06) | 0.78 (0.05) |
| Group DRO | 72.7 (2.1) | 23.9 (2.0) | 68.4 (7.3) | 23.0 (0.3) | 30.8 (0.81) | 52.1 (0.5) | 0.39 (0.06) | 0.75 (0.07) |
| IRM | 59.8 (3.7) | 15.1 (4.9) | 64.2 (8.1) | 8.2 (1.1) | 30.0 (1.37) | 50.8 (0.13) | 0.43 (0.07) | 0.77 (0.05) |
| ARM-CML | 70.5 (0.6) | 28.6 (0.1) | 84.2 (1.4) | 17.3 (1.8) | 27.2 (0.38) | 45.7 (0.28) | 0.37 (0.08) | 0.75 (0.04) |
| ARM-BN | 70.3 (2.4) | 23.7 (2.7) | 87.2 (0.9) | **31.2 (0.1)** | 24.6 (0.04) | 42.0 (0.21) | 0.49 (0.21) | **0.84 (0.05)** |
| ARM-LL | 71.4 (0.6) | 27.4 (0.8) | 84.2 (2.6) | 24.3 (0.3) | 22.1 (0.46) | 42.7 (0.71) | 0.41 (0.04) | 0.76 (0.04) |
| Ours (w/o mask) | 74.1 (0.4) | **35.1 (0.9)** | **90.8 (1.3)** | 29.6 (0.5) | **36.8 (1.01)** | 50.6 (0.20) | **0.52 (0.04)** | 0.80 (0.03) |
| Ours | **77.2 (0.3)** | 34.0 (0.6) | **91.4 (1.5)** | 29.8 (0.4) | **35.4 (0.58)** | 52.5 (0.18) | 0.51 (0.04) | 0.80 (0.03) |

**Network architecture.** We follow WILDS [39] to use ResNet18 & 50 [32] or DenseNet101 [36] for the expert models $\{\mathcal{M}^i\}_{i=1}^N$ and student network $f(\cdot, ; \theta)$. Also, we use a single-layer transformer encoder block[73] as the knowledge aggregator $\mathcal{A}(\cdot; \phi)$. To investigate the resource-constrained and privacy-sensitive scenarios, we utilize MobileNet V2 [61] with a width multiplier of 0.25. As for DomainNet and PACS, we follow the setting in DomainBed to use ResNet50 for both experts and student networks.

**Pre-training domain-specific models.** The WILDS benchmark is highly imbalanced in data size, and some domains might contain empty classes. We found that using every single domain to train an expert is unstable, and sometimes it cannot converge. Inspired by [52], we propose to cluster the training domains into $N$ super domains and use each super-domain to train the expert models. Specifically, we set $N = \{10, 5, 3, 4, 3\}$ for iWildCam, Camelyon17, RxRx1, FMoW and Poverty Map, respectively. We use ImageNet [21] pre-trained model as the initialization and separately train the models using Adam optimizer [38] with a learning rate of $1e^{-4}$ and a decay of 0.96 per epoch.

**Meta-training and testing.** We first pre-train the aggregator and student network [15]. After that, the model is further trained using Alg. 1 for 15 epochs with a fixed learning rate of $3e^{-4}$ for $\alpha$ and $3e^{-5}$ for $\beta$. During meta-testing, we use Line 13 of Alg. 1 to adapt before making a prediction for every testing domain. Specifically, we set the number of examples for adaptation at test time = $\{24, 64, 75, 64, 64\}$ for iWildCam, Camelyon17, RxRx1, FMoW, and Poverty Map, respectively. For both meta-training and testing, we perform one gradient update for adaptation on the unseen target domain. We refer the readers to the supplementary materials for more detailed information.

## 5.2 Main results

**Comparison on WILDS.** We compare the proposed method with prior approaches showing on WILDS leaderboard [39], including non-adaptive methods: CORAL [67], ERM [72], IRM [3], Group DRO [60] and adaptive methods used in ARM [86] (CML, BN and LL). We directly copy the available results from the leaderboard or their corresponding paper. As for the missing ones, we conduct experiments using their provided source code with default hyperparameters. Table 1 reports the comparison with the state-of-the-art. Our proposed method performs well across all datasets and increases both worst-case and average accuracy compared to other methods. Our proposed method achieves the best performance on 4 out of 5 benchmark datasets. ARM [86] applies a meta-learning approach to learn how to adapt to unseen domains with unlabeled data. However, their method is greatly bounded by using a single model to exploit knowledge from multiple source domains. Instead, our proposed method is more fitted to multi-source domain settings and meta-trains an aggregator that properly mixes the knowledge from multiple domain-specific experts. As a result, our method outperforms ARM-CML, BN, and LL by 9.5%, 9.8%, 8.1% for iWildCam, 8.5%, 4.8%, 8.5% for Camelyon17 and 14.8%, 25.0%, 22.9% for FMoW in terms of average accuracy. Furthermore, we also evaluate our method without masking the in-distribution domain in MoE models (Ours w/o mask) during meta-training (Line 10-11 of Alg. 1), where the sampled domain is overlapped with MoE. It violates the generalization to unseen target domains during testing. As most of the performance dropped, it reflects the importance of aligning the training and evaluation objectives.

**Comparison on DomainNet and PACS.** Table 2 and Table 3 report the results on DomainNet and PACS. In DomainNet, our method performs the best on all experimental settings and outperforms

Table 2: Evaluation on DomainNet. Our method performs the best on all experimental settings and outperforms recent SOTA significantly in terms of average accuracy.

| Method | clip | info | paint | quick | real | sketch | avg |
|---|---|---|---|---|---|---|---|
| ERM [72] | 58.1 (0.3) | 18.8 (0.3) | 46.7 (0.3) | 12.2 (0.4) | 59.6 (0.1) | 49.8 (0.4) | 40.9 |
| IRM [3] | 48.5 (2.8) | 15.0 (1.5) | 38.3 (4.3) | 10.9 (0.5) | 48.2 (5.2) | 42.3 (3.1) | 33.9 |
| Group DRO [60] | 47.2 (0.5) | 17.5 (0.4) | 33.8 (0.5) | 9.3 (0.3) | 51.6 (0.4) | 40.1 (0.6) | 33.3 |
| Mixup [77] | 55.7 (0.3) | 18.5 (0.5) | 44.3 (0.5) | 12.5 (0.4) | 55.8 (0.3) | 48.2 (0.5) | 39.2 |
| MLDG [43] | 59.1 (0.2) | 19.1 (0.3) | 45.8 (0.7) | 13.4 (0.3) | 59.6 (0.2) | 50.2 (0.4) | 41.2 |
| CORAL [67] | 59.2 (0.1) | 19.7 (0.2) | 46.6 (0.3) | 13.4 (0.4) | 59.8 (0.2) | 50.1 (0.6) | 41.5 |
| DANN [26] | 53.1 (0.2) | 18.3 (0.1) | 44.2 (0.7) | 11.8 (0.1) | 55.5 (0.4) | 46.8 (0.6) | 38.3 |
| MTL [11] | 57.9 (0.5) | 18.5 (0.4) | 46.0 (0.1) | 12.5 (0.1) | 59.5 (0.3) | 49.2 (0.1) | 40.6 |
| SegNet [55] | 57.7 (0.3) | 19.0 (0.2) | 45.3 (0.3) | 12.7 (0.5) | 58.1 (0.5) | 48.8 (0.2) | 40.3 |
| ARM [86] | 49.7 (0.3) | 16.3 (0.5) | 40.9 (1.1) | 9.4 (0.1) | 53.4 (0.4) | 43.5 (0.4) | 35.5 |
| Ours | **63.5 (0.2)** | **21.4 (0.3)** | **51.3 (0.4)** | **14.3 (0.3)** | **62.3 (1.0)** | **52.4 (0.2)** | **44.2** |

Table 3: Evaluation on PACS. Our method outperforms other methods in 2 out of 4 experiments but still achieves the SOTA in terms of average accuracy.

| Method | art | cartoon | photo | sketch | avg |
|---|---|---|---|---|---|
| ERM [72] | 84.7 (0.4) | 80.8 (0.6) | 97.2 (0.3) | 79.3 (1.0) | 85.5 |
| CORAL [67] | **88.3 (0.2)** | 80.0 (0.5) | **97.5 (0.3)** | 78.8 (1.3) | 86.2 |
| Group DRO [60] | 83.5 (0.9) | 79.1 (0.6) | 96.7 (0.3) | 78.3 (2.0) | 84.4 |
| IRM [3] | 84.8 (1.3) | 76.4 (1.1) | 96.7 (0.6) | 76.1 (1.0) | 83.5 |
| ARM [86] | 86.8 (0.6) | 76.8 (0.5) | 97.4 (0.3) | 79.3 (1.2) | 85.1 |
| Ours | 86.1 (0.2) | **82.5 (0.5)** | 96.7 (0.4) | **82.3 (1.4)** | **86.9** |

recent SOTA significantly in terms of the average accuracy (+2.7). [82] has discovered that the lack of a large number of meta-training episodes leads to the meta-level overfitting/memorization problem. To our task, since PACS has 57× less number of images than DomainNet and 80× less number of domains than iWildCam, the capability of our meta-learning-based method is hampered by the less diversity of episodes. As a result, we outperform other methods in 2 out of 4 experiments but still achieve the SOTA in terms of average accuracy.

**Visualization of adapted features.** To evaluate the capability of adaptation via learning discriminative representations on unseen target domains, we compare the t-SNE [71] feature visualization using the same set of test domains sampled from iWildCam and Camelyon17 datasets. ERM utilizes a single model and standard supervised training without adaptation. Therefore, we set it as the baseline. Figure 2 shows the comparison, where each color denotes a class, and each point represents a data sample. It is clear that our method obtains better clustered and more discriminative features.

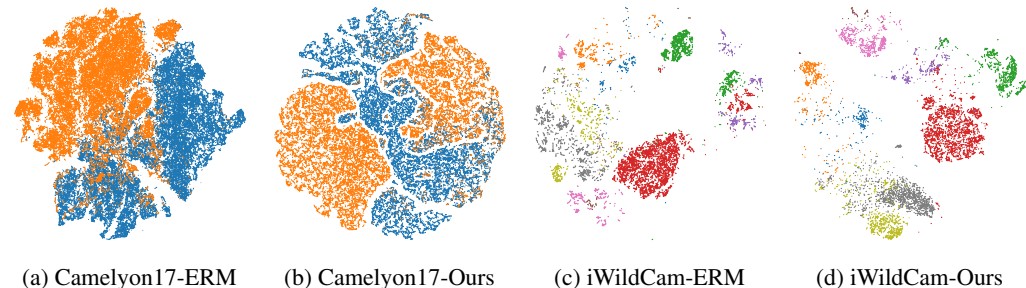

(a) Camelyon17-ERM     (b) Camelyon17-Ours     (c) iWildCam-ERM     (d) iWildCam-Ours

Figure 2: t-SNE visualization of adapted features at test-time. We directly utilize features adapted to the same unseen target domains from ERM and our proposed method in Camelyon17 and iWildCam datasets, respectively. Our resulting features show more discriminative decision boundaries.

## 5.3 Results under constrained real-world settings

In this section, we mainly conduct experiments on iWildCam dataset under two real-world settings.

**Constraint on computational cost.** Computational power is always limited in real-world deployment scenarios, such as edge devices. Efficiency and adaptation ability should both be considered. Thus, we replace our student model and the models in other methods with MobileNet V2. As reported

Table 4: Comparison of WILDS testbeds using MobileNet V2. Reducing the model size hampers the learning capability. Our method shows a better trade-off as the knowledge is distilled from MoE.

| | iWildCam | | Camelyon17 | RxRx1 | FMoW | | PovertyMap | |
|---|---|---|---|---|---|---|---|---|
| Method | Acc | Macro F1 | Acc | Acc | WC Acc | Avg Acc | WC Pearson r | Pearson r |
| ERM | 56.7 (0.7) | 17.5 (1.2) | 69.0 (8.8) | 14.3 (0.2) | 15.7 (0.68) | **40.0 (0.11)** | 0.39 (0.05) | 0.77 (0.04) |
| CORAL | **61.5 (1.7)** | 17.6 (0.1) | 75.9 (6.9) | 12.6 (0.1) | 22.7 (0.76) | 31.0 (0.32) | **0.44 (0.06)** | **0.79 (0.04)** |
| ARM-CML | 58.2 (0.8) | 15.8 (0.6) | 74.9 (4.6) | 14.0 (1.4) | 21.1 (0.33) | 30.0 (0.13) | 0.41 (0.05) | 0.76 (0.03) |
| ARM-BN | 54.8 (0.6) | 13.8 (0.2) | 85.6 (1.6) | 14.9 (0.1) | 17.9 (1.82) | 29.0 (0.69) | 0.42 (0.05) | 0.76 (0.03) |
| ARM-LL | 57.5 (0.5) | 12.6 (0.8) | 84.8 (1.7) | 15.0 (0.2) | 17.1 (0.22) | 30.3 (0.54) | 0.39 (0.07) | 0.76 (0.02) |
| Ours | 59.5 (0.7) | **19.7 (0.5)** | **87.1 (2.3)** | 15.1 (0.4) | **26.9 (0.67)** | 37.9 (0.31) | **0.44 (0.04)** | 0.77 (0.03) |

in Table 4, our proposed method still outperforms prior methods. Since the MoE model is only used for knowledge transfer, our method is more flexible in designing the student architecture for different scenarios. We also report multiply-Accumulate operations (MACS) for inference and time complexity on adaptation. As ARM needs to make adaptations before inference on every example, its adaptation cost scales linearly with the number of examples. Our proposed method performs better in accuracy and requires much less computational cost for adaptation, as reported in Table 5.

**Constraint on data privacy.** On top of computational limitations, privacy-regulated scenarios are common in the real world. It introduces new challenges as the raw data is inaccessible. Our method does not need to access the raw data but the trained models, which greatly mitigates such regulation. Thus, as shown in Table 6, our method does not suffer from much performance degradation compared to other methods that require access to the private raw data.

Table 5: Adaptation efficiency evaluated on iWild-Cam using MobileNet V2. Our method not only outperforms prior methods but also keeps constant time complexity in test-time adaptation.

| Method | Acc / Macro-F1 | MACS | Complexity |
|---|---|---|---|
| ERM | 56.7 / 17.5 | $7.18 \times 10^7$ | N/A |
| ARM-CML | 58.2 / 15.8 | $7.73 \times 10^7$ | $O(n)$ |
| ARM-LL | 57.5 / 12.6 | $7.18 \times 10^7$ | $O(n)$ |
| Ours | 59.5 / 19.7 | $7.18 \times 10^7$ | $O(1)$ |

### 5.4 Ablation studies

In this section, we conduct ablation studies on iWildCam to analyze various components of the proposed method. We also seek to answer the two key questions: 1) Does the number of experts affect the capability of capturing knowledge from multi-source domains? 2) Is meta-learning superior to standard supervised learning under the knowledge distillation framework?

**Number of domain-specific experts.** We investigate the impact of exploiting multiple experts to store domain-specific knowledge separately. Specifically, we keep the total number of data for experts' pretraining fixed and report the results using a various number of expert models. The experiments in Table 7 validate the benefits of using more domain-specific experts.

**Training scheme.** To verify the effectiveness of meta-learning, we investigate three training schemes: random initialization, pre-train, and meta-train. To pre-train the aggregator, we add a classifier layer to its aggregated output and follow the standard supervised training scheme.

Table 6: Results on privacy-related regulation setting evaluated on iWildCam and FMoW using MobileNet V2. Without privacy considered in the design, prior methods can only exploit public data and thus achieve far worse performance.

| | iWildCam | | FMoW | |
|---|---|---|---|---|
| Method | Acc | Macro-F1 | WC Acc | Acc |
| ERM | 51.2 | 11.2 | 22.5 | **35.4** |
| CORAL | 50.2 | 11.1 | 18.1 | 25.4 |
| ARM-CML | 42.7 | 7.5 | 16.8 | 24.1 |
| ARM-BN | 46.9 | 8.7 | 14.2 | 22.2 |
| ARM-LL | 46.8 | 9.3 | 13.7 | 22.6 |
| Ours | **54.7** | **14.2** | **24.4** | 33.8 |

Table 7: Results on different numbers of domain-specific experts. More experts increase the learning capacity to better explore each source domain, thus, improving generalization.

| # of experts | 2 | 5 | 7 | 10 |
|---|---|---|---|---|
| Accuracy | 70.4 | 74.1 | 76.4 | 77.2 |
| Macro-F1 | 30.6 | 32.3 | 33.7 | 34.0 |

For fair comparisons, we use the same testing scheme, including the number of updates and images for adaptation. Table 8 reports the results of different training scheme combinations. We observe that the randomly initialized student model struggles to learn from few-shot data. And the pre-trained aggregator brings weaker adaptation guidance to the student network as the aggregator is not learned to distill. In contrast, our bi-level optimization-based training scheme enforces the aggregator to choose more correlated knowledge

from multiple experts to improve the adaptation of the student model. Therefore, the meta-learned aggregator is more optimal (row 1 vs. row 2). Furthermore, our meta-distillation training process simulates the adaptation in testing scenarios, which aligns with the training objective and evaluation protocol. Hence, for both meta-trained aggregator and student models, it gains additional improvement (row 3 vs. row 4).

**Aggregator and distillation methods.** Table 9 reports the effects of various aggregators, including two hand-designed operators: Max and Average pooling, and two MLP-based methods: Weighted sum (MLP-WS) and Projector (MLP-P) (details are provided in the supplementary materials). We found that the fully learned transformer-based aggregator is crucial for mixing domain-specific features. Another important design choice in our proposed framework is the form of knowledge: distilling the teacher model's logits, intermediate features, or both. We show the evaluation results of those three forms of knowledge in Table 10.

Table 8: Evaluation of training schemes. Using both meta-learned aggregator and student model improves generalization as they are learned towards test-time adaptation.

| Train Scheme | | Metrics | |
| --- | --- | --- | --- |
| Aggregator | Student | Acc | Macro-F1 |
| Pretrain | Random | 6.2 | 0.1 |
| Meta | Random | 32.7 | 0.5 |
| Pretrain | Meta | 74.8 | 32.9 |
| Meta | Meta | 77.2 | 34.0 |

Table 9: Comparison between different aggregator methods. The transformer explores interconnection, which gives the best result.

| | Max | Ave. | MLP-WS | MLP-P | Trans.(ours) |
| --- | --- | --- | --- | --- | --- |
| Acc. | 69.2 | 69.7 | 70.7 | 73.7 | 77.2 |
| M-F1 | 29.2 | 25.0 | 32.8 | 32.7 | 34.0 |

Table 10: Comparison between different distillation methods. Distilling only the feature extractor yields the best generalization.

| | Logits | Logits + Feat. | Feat. (Ours) |
| --- | --- | --- | --- |
| Accuracy | 72.1 | 73.1 | 77.2 |
| Marco-F1 | 26.4 | 26.9 | 34.0 |

## 6 Discussion

We present Meta-DMoE, a framework for adaptation towards domain shift using unlabeled examples at test-time. We formulate the adaptation as a knowledge distillation process and devise a meta-learning algorithm to guide the student network to fast adapt to unseen target domains via transferring the aggregated knowledge from multi-source domain-specific models. We demonstrate that Meta-DMoE is state-of-the-art on four benchmarks. And it is competitive under two constrained real-world settings, including limited computational budget and data privacy considerations.

**Limitations.** As discussed in Section 5.4, Meta-DMoE can improve the capacity to capture complex knowledge from multi-source domains by increasing the number of experts. However, to compute the aggregated knowledge from domain-specific experts, every expert model needs to have one feed-forward pass. As a result, the computational cost of adaptation scales linearly with the number of experts. Furthermore, to add or remove any domain-specific expert, both the aggregator and the student network need to be re-trained. Thus, enabling a sparse-gated Meta-DMoE to encourage efficiency and scalability could be a valuable future direction, where a gating module determines a sparse combination of domain-specific experts to be used for each target domain.

**Social impact.** Tackling domain shift problems can have positive social impacts as it helps to elevate the model accuracy in real-world scenarios (e.g., healthcare and self-driving cars). In healthcare, domain shift occurs when a trained model is applied to patients in different hospitals. In this case, model performance might dramatically decrease, which leads to severe consequences. Tackling domain shifts helps to ensure that models can work well on new data, which can ultimately lead to better patient care. We believe our work is a small step toward the goal of adapting to domain shift.

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
