# A  Additional Ablation Studies

In this section, we provide three additional ablation studies and discussions to further analyze our proposed method. These ablation studies are conducted on the iWildCam dataset.

## A.1  Aggregator Methods

In Table 9, we include several hand-designed aggregation operators: max-pooling, average-pooling, and two MLP-based learnable architectures. The two MLP-based learnable architectures work as follows.

MLP weighted sum (MLP-WS) takes the output features from the MoE models as input and produces the score for each expert. Then, we weigh those output features using the scores and sum them to obtain the final output for knowledge distillation.

For the MLP projector (MLP-P), the output features from the MoE are flattened at first ($N \times D \to ND \times 1$) and then fed into an MLP architecture ($ND \times D, D \times D$) to obtain the final output ($D \times 1$) for knowledge distillation.

## A.2  Excluding Overlapping Expert

As discussed in Section 4.1, we simulate the test-time out-of-distribution by excluding the corresponding expert model in each episode since the training domains overlap for the MoE and meta-training. If the corresponding expert model is not excluded during meta-training, the aggregator output might be dominated by the corresponding expert output, or even collapse into a domain classification problem from the perspective of the aggregator. This might hinder the generalization on OOD domains. The experiments in Table 11 also validate the benefits of using such an operation.

## A.3  Expert Architecture

In this section, we analyze the effects of using a different expert architecture. Table 12 validates the benefits of using the knowledge aggregator and our proposed training algorithm. Our proposed method could perform robustly across different expert architectures.

Table 11: Comparison using ID test split in iWildCam. The ID test split contains images from the same domains as the training set but on different days from the training images. The model trained without masks performs better than the model trained with masks under the ID test split but has lower accuracy and a comparable Macro-F1 than the model trained with masks in the OOD test split.

| MoE Mask | ID Acc | ID Macro-F1 | OOD Acc | OOD Macro-F1 |
|---|---|---|---|---|
| Mask all except overlap | 75.5 | 46.8 | —- | —- |
| Without mask | 76.4 | 48.0 | 74.1 | 35.1 |
| With mask | 72.9 | 44.4 | 77.2 | 34.0 |

Table 12: Comparison with different expert architectures. Our proposed method is robust to different expert architectures with different capacities.

| Expert architecture | Student architecture | Acc | Macro-F1 |
|---|---|---|---|
| MobileNet V2 | MobileNet V2 | 59.5 | 19.7 |
| ResNet-50 | MobileNet V2 | 58.8 | 21.0 |

## A.4  Number of Images Used for Test-Time Adaptation

During deployment, our method uses a small number of unlabelled images to adapt the student prediction network to the target domain. Increasing the number of images used for adaptation might give a better approximation of the marginal of the target domain. Thus, the performance in the target domains is also enhanced. The experiments in Table 13 validate the benefits of using more images for adaptation.

Table 13: Results on the number of images for adaptation. Adaptation using more images leads to better approximations of the marginal and improves generalization.

| # of images for adaptation | 2 | 4 | 8 | 16 | 24 |
|---|---|---|---|---|---|
| Accuracy | 76.5 | 76.9 | 77.0 | 77.2 | 77.2 |
| Macro-F1 | 31.5 | 31.2 | 31.7 | 33.0 | 34.0 |

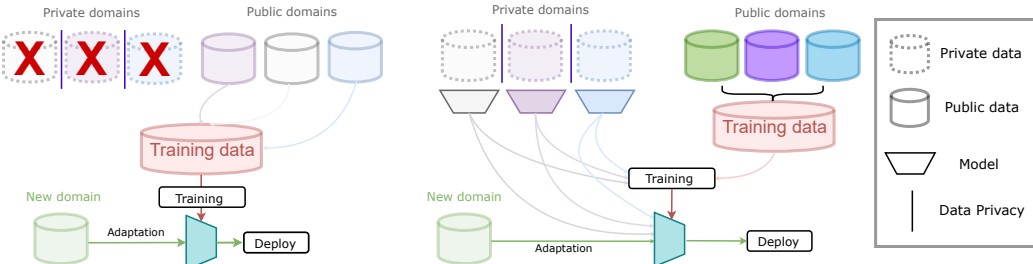

Figure 3: Left: Standard methods require sampling mini-batched data across domains and thus cannot utilize the locally-stored private data within each private domain. Right: Privacy-related algorithms can improve the adaptation results by transferring knowledge from the private data without access to the raw data.

## B  Details on Privacy Constrained Setting

### B.1  Problem Definition

In this section, we discuss a problem setting where data privacy regulation is imposed. To achieve data diversity, large-scale labeled training data are normally collected from public venues (internet or among institutes) and stored in a server where i.i.d conditions can be satisfied to train a generic model by sampling mini-batches. However, in real-world applications, due to privacy-related regulations, some datasets cannot be shared among users or distributed edges. Such data can only be processed locally. Thus, they cannot be directly used for training a generalized model in most existing approaches [24, 51]. In this work, we consider a more realistic deployment problem with privacy constraints imposed.

We illustrate the privacy-regulated test-time adaptation setting in Fig. 3. To simulate the privacy-regulated scenario, we explicitly separate the distributed training source domains into two non-overlapping sets of domains: $\mathcal{D_S}^{pri}$ for private domains and $\mathcal{D_S}^{pub}$ for public domains. Each domain within $\mathcal{D_S}^{pri}$ contains private data that can only be shared and accessed within that domain. Therefore, the data within $\mathcal{D_S}^{pri}$ can only be accessed locally in a distributed manner during training, and cannot be seen at test time. $\mathcal{D_S}^{pub}$ contains domains with only public data that has fewer restrictions and can be accessed from a centralized platform. Such splitting allows the use of $\mathcal{D_S}^{pub}$ to simulate $\mathcal{D_T}$ at training to learn the interaction with $\mathcal{D_S}^{pri}$. It is also possible for some algorithms to mix all $\mathcal{D_S}^{pub}$ and store them in a server to draw a mini-batch for every training iterations [67, 3], but such operation is not allowed for private data.

The ultimate goal under this privacy-regulated setting is to train a recognition model on domains $\mathcal{D_S}^{pri}$ and $\mathcal{D_S}^{pub}$ with the above privacy regulations applied. The model should perform well in the target domains $\mathcal{D_T}$ without accessing either $\mathcal{D_S}^{pri}$ or $\mathcal{D_S}^{pub}$.

### B.2  Applying Meta-DMoE to Privacy Constrained Setting

Our proposed Meta-DMoE method is a natural solution to this setting. Concretely, for each private domain $\mathcal{D_S}^{i,pri}$, we train an expert model $\mathcal{M}_e^i$ using only data from $\mathcal{D_S}^{i,pri}$. After obtaining the domain-specific experts $\{\mathcal{M}_e^i\}$, we perform the subsequent meta-training on $\mathcal{D_S}^{pub}$ to simulation OOD test-time adaptation. The training algorithm is identical to Alg. 1, except we don't mask any experts' output since the training domains for the MoEs and meta-training do not overlap. In this

way, we can leverage the knowledge residing in $\mathcal{D_S}^{pri}$ without accessing the raw data but only the trained model on each domain during centralized meta-training. We also include the details of the experiments under this setting in Appendix D.1.

## C  Details on Knowledge Aggregator

In this section, we discuss the detailed architecture and computation of the knowledge aggregator. We use a naive single-layer transformer encoder [73, 22] to implement the aggregator. The transformer encoder consists of multi-head self-attention blocks (MSA) and multi-layer perceptron blocks (MLP) with layernorm (LN) [4] applied before each block, and residual connection applied after each block. Formally, given the concatenated output features from the MoE models,

$$\mathbf{z}_0 = Concat[\mathcal{M}_e^1(\mathbf{x}), \mathcal{M}_e^2(\mathbf{x}), ..., \mathcal{M}_e^N(\mathbf{x})] \in \mathbb{R}^{N \times d}, \tag{4}$$

$$\mathbf{z}_0' = MSA_k(LN(\mathbf{z}_0)) + \mathbf{z}_0, \tag{5}$$

$$\mathbf{z}_{out} = MLP(LN(\mathbf{z}_0')) + \mathbf{z}_0', \tag{6}$$

where $MSA_k(\cdot)$ is the MSA block with $k$ heads and a head dimension of $d_k$ (typically set to $d/k$),

$$[\mathbf{q}, \mathbf{k}, \mathbf{v}] = \mathbf{z}\mathbf{W}_{qkv} \quad \mathbf{W}_{qkv} \in \mathbb{R}^{d \times 3 \cdot d_k}, \tag{7}$$

$$SA(\mathbf{z}) = Softmax(\frac{\mathbf{q}\mathbf{k}^T}{\sqrt{d_k}})\mathbf{v}, \tag{8}$$

$$MSA_k(\mathbf{z}) = Concat[SA_1(\mathbf{z}), ..., SA_k(\mathbf{z})]\mathbf{W}_o \quad \mathbf{W}_o \in \mathbb{R}^{k \cdot D_k \times D} . \tag{9}$$

We finally average-pool the transformer encoder output $\mathbf{z}_{out} \in \mathbb{R}^{N \times d}$ along the first dimension to obtain the final output. In the case when the dimensions of the features outputted by the aggregator and the student are different, we apply an additional MLP layer with layernorm on $\mathbf{z}_{out}$ to reduce the dimensionality as desired.

## D  Additional Experimental Details

We run all the experiments using a single NVIDIA V100 GPU. The official WILDS dataset contains training, validation, and testing domains which we use as source, validation target, and test target domains. The validation set in WILDS [39] contains held-out domains with labeled data that are non-overlapping with training and testing domains. To be specific, we first use the training domains to pre-train expert models and meta-train the aggregator and the student prediction model and then use the validation set to tune the hyperparameters of meta-learning. At last, we evaluate our method with the test set. We include the official train/val/test domain split in the following subsections. We run each experiment and report the average as well as the unbiased standard deviation across three random seeds unless otherwise noted. In the following subsections, we provide the hyperparameters and training details for each dataset below. For all experiments, we select the hyperparameters settings using the validation split on the default evaluation metrics from WILDS. For both meta-training and testing, we perform one gradient update for adaptation on the unseen target domain.

### D.1  Details for Privacy Constrained Evaluation

We mainly perform experiments under privacy constrained setting on two subsets of WILDS for image recognition tasks, iWildCam and FMoW. To simulate the privacy constrained scenarios, we randomly select 100 domains from iWildCam training split as $\mathcal{D_S}^{pri}$ to train $\{\mathcal{M}_e^i\}_{i=1}^M$ and the rest as $\mathcal{D_S}^{pub}$ to meta-train the knowledge aggregator and student network. As for FMoW, we randomly select data from 6 years as $\mathcal{D_S}^{pri}$ and the rest as $\mathcal{D_S}^{pub}$. The domains are merged into 10 and 3 super-domains, respectively, as discussed in Section 5.1. Since ARM and other methods only utilize the data as input, we train them on only $\mathcal{D_S}^{pub}$.

## D.2 IWildCam Details

IWildCam is a multi-class species classification dataset, where the input $x$ is an RGB photo taken by a camera trap, the label $y$ indicates one of 182 animal species, and the domain $z$ is the ID of the camera trap. During training and testing, the input $x$ is resized to $448 \times 448$. The train/val/test set contains 243/32/48 domains, respectively.

**Evaluation.** Models are evaluated on the Macro-F1 score, which is the F1 score across all classes. According to [39], Macro-F1 score might better describe the performance on this dataset as the classes are highly imbalanced. We also report the average accuracy across all test images.

**Training domain-specific model.** For this dataset, we train 10 expert models where each expert is trained on a super-domain formed by 24-25 domains. The expert model is trained using a ResNet-50 model pretrained on ImageNet. We train the expert models for 12 epochs with a batch size of 16. We use Adam optimizer with a learning rate of 3e-5.

**Meta-training and testing.** We train the knowledge aggregator using a single-layer transformer encoder with 16 heads. The transformer encoder has an input and output dimension of 2048, and the inner layer has a dimension of 4096. We use ResNet-50 [32] model for producing the results in Table 1. We first train the aggregator and student network with ERM until convergence for faster convergence speed during meta-training. After that, the models are meta-trained using Alg. 1 with a learning rate of 3e-4 for $\alpha$, 3e-5 for $\beta_s$, 1e-6 for $\beta_a$ using Adam optimizer, and decay of 0.98 per epoch. Note that we use a different meta learning rate, $\beta_a$ and $\beta_s$ respectively, for the knowledge aggregator and the student network as we found it more stable during meta training. In each episode, we first uniformly sample a domain, and then use 24 images in this domain for adaptation and use 16 images to query the loss for meta-update. We train the models for 15 epochs with early stopping on validation Macro-F1. During testing, we use 24 images to adapt the student model to each domain.

## D.3 Camelyon Details

This dataset contains 450,000 lymph node scan patches extracted from 50 whole-slide images (WSIs) with 10 WSIs from each of 5 hospitals. The task is to perform binary classification to predict whether a region of tissue contains tumor tissue. Under this task specification, the input $x$ is a 96 by 96 scan patch, the label $y$ indicates whether the central region of a patch contains tumor tissue, and the domain $z$ identifies the hospital. The train/val/test set contains 30/10/10 WSIs, respectively.

**Evaluation.** Models are evaluated on the average accuracy across all test images.

**Training domain-specific model.** For this dataset, we train 5 expert models where each expert is trained on a super-domain formed by 6 WSIs since there are only 3 hospitals in the training split. The expert model is trained using a DenseNet-121 model from scratch. We train the expert models for 5 epochs with a batch size of 32. We use an Adam optimizer with a learning rate of 1e-3 and an $L2$ regularization of 1e-2.

**Meta-training and testing.** We train the knowledge aggregator using a single-layer transformer encoder with 16 heads. The knowledge aggregator has an input and output dimension of 1024, and the inner layer has a dimension of 2048. We use DenseNet-121 [36] model for producing the results in Table 1. We first train the aggregator until convergence, and the student network is trained from ImageNet pretrained. After that, the models are meta-trained using Alg. 1 with a learning rate of 1e-3 for $\alpha$, 1e-4 for $\beta_s$, 1e-3 for $\beta_a$ using Adam optimizer and a decay of 0.98 per epoch for 10 epochs. In each episode, we first uniformly sample a WSI, and then use 64 images in this WSI for adaptation and use 32 images to query the loss for meta-update. The model is trained for 10 epochs with early stopping. During testing, we use 64 images to adapt the student model to each WSI.

## D.4 RxRx1 Details

The task is to predict 1 of 1,139 genetic treatments that cells received using fluorescent microscopy images of human cells. The input $x$ is a 3-channel fluorescent microscopy image, the label $y$ indicates which of the treatments the cells received, and the domain $z$ identifies the experimental batch of the image. The train/val/test set contains 33/4/14 domains, respectively.

**Evaluation.** Models are evaluated on the average accuracy across all test images.

**Training domain-specific model.** For this dataset, we train 3 expert models where each expert is trained on a super-domain formed by 11 experiments. The expert model is trained using a ResNet-50 model pretrained from ImageNet. We train the expert models for 90 epochs with a batch size of 75. We use an Adam optimizer with a learning rate of 1e-4 and an $L2$ regularization of 1e-5. We follow [39] to linearly increase the learning rate for the first 10 epochs and then decrease it using a cosine learning rate scheduler.

**Meta-training and testing.** We train the knowledge aggregator using a single-layer transformer encoder with 16 heads. The knowledge aggregator has an input and output dimension of 2048, and the inner layer has a dimension of 4096. We use the ResNet-50 model to produce the results in Table 1. We first train the aggregator and student network with ERM until convergence. After that, the models are meta-trained using Alg. 1 with a learning rate of 1e-4 for $\alpha$, 1e-6 for $\beta_s$, 3e-6 for $\beta_a$ using Adam optimizer and following the cosine learning rate schedule for 10 epochs. In each episode, we use 75 images from the same domain for adaptation and use 48 images to query the loss for meta-update. During testing, we use 75 images to adapt the student model to each domain.

### D.5    FMoW Details

FMoW is comprised of high-definition satellite images from over 200 countries based on the functional purpose of land in the image. The task is to predict the functional purpose of the land captured in the image out of 62 categories. The input $x$ is an RBD satellite image resized to $224 \times 224$, the label $y$ indicates which of the categories that the land belongs to, and the domain $z$ identifies both the continent and the year that the image was taken. The train/val/test set contains 55/15/15 domains, respectively.

**Evaluation.** Models are evaluated by the average accuracy and worst-case (WC) accuracy based on geographical regions.

**Training domain-specific model.** For this dataset, we train 4 expert models where each expert is trained on a super-domain formed by all the images in 2-3 years. The expert model is trained using a DenseNet-121 model pretrained from ImageNet. We train the expert models for 20 epochs with a batch size of 64. We use an Adam optimizer with a learning rate of 1e-4 and a decay of 0.96 per epoch.

**Meta-training and testing.** We train the knowledge aggregator using a single-layer transformer encoder with 16 heads. The knowledge aggregator has an input and output dimension of 1024, and the inner layer has a dimension of 2048. We use the DenseNet-121 model to produce the results in Table 1. We first train the aggregator and student network with ERM until convergence. After that, the models are meta-trained using Alg. 1 with a learning rate of 1e-4 for $\alpha$, 1e-5 for $\beta_s$, 1e-6 for $\beta_a$ using Adam optimizer and a decay of 0.96 per epoch. In each episode, we first uniformly sample a domain from {continent $\times$ year}, and then use 64 images from this domain for adaptation and use 48 images to query the loss for meta-update. We train the models for 30 epochs with early stopping on validation WC accuracy. During testing, we use 64 images to adapt the student model to each domain.

### D.6    Poverty Details

The task is to predict the real-valued asset wealth index using a multispectral satellite image. The input $x$ is an 8-channel satellite image resized to $224 \times 224$, the label $y$ is a real-valued asset wealth index of the captured location, and the domain $z$ identifies both the country that the image was taken and whether the area is urban or rural. For this dataset, we use MSE Loss for training the domain-specific experts and meta-training. The train/val/test set contains 26-28/8-10/8-10 domains, respectively. The number of domains varies slightly from the fold to the fold for Poverty.

**Evaluation.** Models are evaluated by the Pearson correlation (r) and worst-case (WC) r based on urban/rural sub-populations. This dataset is split into 5 folds where each fold defines a different set of Out-of-Distribution (OOD) countries. The results are aggregated over 5 folds.

**Training domain-specific model.** For this dataset, we train 3 expert models where each expert is trained on a super-domain formed by 4-5 countries. The expert model is trained using a ResNet-18 model from scratch. We train the expert models for 70 epochs with a batch size of 64. We use an Adam optimizer with a learning rate of 1e-3 and a decay of 0.96 per epoch.

**Meta-training and testing.** We train the knowledge aggregator using a single-layer transformer encoder with 16 heads. The knowledge aggregator has an input and output dimension of 512, and the inner layer has a dimension of 1024. We use the ResNet-18 model to produce the results in Table 1. We first train the aggregator and student network with ERM until convergence. After that, the models are meta-trained using Alg. 1 with a learning rate of 1e-3 for $\alpha$, 1e-4 for $\beta_s$, 1e-4 for $\beta_a$ using Adam optimizer and a decay of 0.96 per epoch. In each episode, we first uniformly sample a domain from {country $\times$ urban/rural}, and then use 64 images from this domain for adaptation and use 64 images to query the loss for meta-update. We train the models for 100 epochs with early stopping on validation Pearson r. During testing, we use 64 images to adapt the student model to each domain.