# OpenReview forum: "Meta-DMoE: Adapting to Domain Shift by Meta-Distillation from Mixture-of-Experts"
_NeurIPS.cc/2022/Conference — NeurIPS 2022 Accept_

### Official Review · Reviewer_RHgP · 2022-07-08

**Rating:** 6
**Confidence:** 2
**Soundness:** 3 good
**Presentation:** 3 good
**Contribution:** 3 good

**Summary:**


Here, the authors present a novel framework (Meta-DMoE) to alleviate domain shift using unlabelled data during test time. Many techniques were combined such as knowledge distillation and meta-learning in order to address existing limitations in previous methods, including training on multiple source domains with a single model and applying it to all subsequent unseen target domains and its bias toward domain-invariant features. The method was evaluated using five test datasets from WILDS  and compared against existing state-of-the-art methods addressing domain shift.


**Questions:**

As stated earlier, please double-check points in the text that might sound a bit repetitive.

**Limitations:**

Limitations were properly addressed in the text.

**Strengths And Weaknesses:**

Strengths

1) Meta-DMoE explores and takes advantage of knowledge distillation and meta-learning to devise a method that can deal with domain shit at test time using unlabelled data. A model with such skill is valuable where training might be expensive and this contribution is important.

2) The authors conducted extensive experiments (WILDS image testbeds and out-of-distribution settings, constraints on computational cost, and privacy regulation ) and Meta-DMoE presented the best results in most cases.

3) Ablation studies were also extensive exploring reasonable conditions.


Weaknesses

Minor:

1) I thought the reading was a bit repetitive, e.g.
Line 70: We name our method as Meta-Distillation of MoE (Meta-DMoE)
Line 162: We name our approach Meta-distillation from MoE (Meta-DMoE)

---

> ### Author Response · Authors · 2022-08-02
> **Author Response to Reviewer RHgP**
>
> We would like to thank the reviewer for taking the time on reviewing our paper and providing valuable feedback.
>
> Thanks for the commendations for the soundness, presentation, and contribution of our work. Regarding the repetitive text, we will conduct additional rounds of proofreading and remove the repetitive text!

---

### Official Review · Reviewer_cPHa · 2022-07-10

**Rating:** 6
**Confidence:** 4
**Soundness:** 3 good
**Presentation:** 2 fair
**Contribution:** 3 good

**Summary:**

The paper tackles the unsupervised domain adaptation scenario, with multiple labeled source domains and an unlabeled target domain. The work draws inspiration from prior work on test-time adaptation, in particular ARM [72]. The key idea is to train a separate model for each source domain, and leverage the mixture of experts (MoE) to fully benefit from the knowledge from these domains. A meta-learning framework is proposed, where a student network aims to distill knowledge from the teacher MoE model. A transformer-based aggregator network is used to combine and contextualize MoE knowledge. Both the teacher and student models are meta-trained in a bi-level optimization framework. The adapted student network is then used during inference. The paper presents experiments on the popular WILDS benchmark, demonstrating competitive performance vs. state-of-the-art methods on a range of tasks.

**Questions:**

- Please address the question above about the number of target samples used by the proposed method vs. the baselines.

- Please address the questions above about the experimental details such as number of target domains and val/test splits.

- Section 4.1: It would be good to clarify, is the source domain data used to train the expert model overlapping or is it disjoint from the support/query sets used later in meta-training? (I would expect that these are not seen while training the expert model?)

- The ablation study "Number of domain-specific experts" L340 needs a clarification, what does "increasing the number of experts" really mean? Is the overall training set fixed and only the "clusters" within it are more coarse/fine? Or is each case using more/less data? (I would expect the former?)

- For the aggregator architecture ablation, it would have been interesting to also include another learned yet non-transformer design (in addition to the naive max/mean).

- Table 8 shows that distilling features alone is better than also distilling the logits, do you have any thoughts on the reasons behind it?

**Limitations:**

The guidelines explicitly state that "The page limit was increased to ensure that authors have space to address the checklist questions." Thus it would have been more appropriate to include the Discussion on Social Impact in the main paper rather than the supplemental.

**Strengths And Weaknesses:**

Strengths
- The proposed approach builds upon related prior work yet appears to be sufficiently novel and also effective.
- Comparison to SoTA seems comprehensive. The proposed method does not outperform SoTA in all cases but delivers strong performance across the board, which is not true about the other methods.
- The authors present a detailed and informative ablation study demonstrating that the choices made in the model lead to superior performance.
- I also appreciate the additional studies concerning the constrained computational cost and data privacy settings.

Weaknesses

Most of the key weaknesses are around presentation/clarity and lack of certain experimental details. (Additional issues are listed under Questions.)

- One of the arguments in favor of Domain Generalization is that "the data samples from a target domain ... might be unavailable" L35. This is emphasized by the authors, but later is ignored, since they operate in the setting where unlabeled target data is available. It is indeed a realistic case that _no target domain data is available_, and I was hoping the authors plan to address that, but that was not the case. Perhaps the presentation could be made clearer on that matter.

- Moreover, it would be good to detail, how much target data is indeed used by the proposed method? From the supplemental material Tab 11 it looks like for iWildCam there are 24 images used for adaptation. Would have been good to include these details (for all the datasets) in the main paper!
- Finally, what about the competing baselines, how many target images were used in those? I.e. is the comparison fair?

- The experimental settings should be detailed more, e.g. we only see the number of source domains in each dataset, but not the number of target domains!
- Moreover, it is unclear how the validation/test splits are defined, e.g. is validation set part of the target domain? Is it labeled or unlabeled? (Perhaps these are standard settings for WILDS, but would be good to clarify for the readers not closely familiar with WILDS!)
- Since it is not stated clearly, is it correct that all the labels are shared across the source and target domains in this work?

Writing
- There are quite a few sentences with incorrect grammar (in the Intro and throughout the paper), I recommend proof-reading.
- L70 "Our contributions are manifold" : I do not think this is the right use of the word "manifold".
- In the Related Work section, ideally each paragraph should include direct contrast of prior works to the proposed method, to make the differences clear.

---

> ### Author Response · Authors · 2022-08-02
> **Author Response to Reviewer cPHa (Part 2/2)**
>
> > *Q6. Is the source domain data used for training expert models overlapped with those used for meta-training (support/query sets) in the aggregator? I would expect that examples for the meta-training are unseen when training the expert model?*
>
> Yes, they are overlapping. However, using the disjoint datasets is not the only way to mimic the out-of-distribution (OOD) scenario. Instead, we introduced a masking mechanism, described in L214 - 217 and B.1 in the supplementary document, to explicitly simulate out-of-distribution cases during meta-training. In short, during meta training, we mask out the expert model corresponding to the sampled domain during the episode. For example, there are three domains {A, B, C} in the training set. During meta training, if domain A is sampled in one episode, we will mask out the features from experts trained using data from A and thus enforce the knowledge distillation from experts trained using data from B and C. Also, we have verified the effectiveness of the masking mechanism on OOD scenarios via the comparison with the result without the mask in Table 1 in the main paper and in Table 9 in the supplementary.
>
> We didn’t directly follow the non-overlapping setting because that setting would decrease the amount of data for meta-training. As a result, sampling less diverse episodes would hamper the effectiveness of meta-learning based methods [79]
>
> > *Q7.Can you clarify the meaning of increasing the number of experts in ABLATION STUDY?*
>
> To make the experiment results faithful, we keep the total number of data for experts pretraining fixed. As the number of experts increases, the number of data used for each expert model will decrease.
>
> > *Q8. For the aggregator architecture ablation, it would have been interesting to also include another learned yet non-transformer design.*
>
> Excellent idea. Thanks for the suggestion. We designed two extra MLP-based learnable architectures and evaluate them following the same setting in Table 7 (both methods have two layers):
>
> * MLP weighted sum (MLP-WS): it takes the output features from the MoE models as input and produces the score for each expert. Then, we weight those output features using the scores and sum them to obtain the final output for knowledge distillation.
> * MLP projector (MLP-P): The output features from the MoE are flattened at first (N*D -> ND * 1) and then fed into an MLP architecture (ND * D, D * D) to obtain the final output (D * 1)  for knowledge distillation.
>
> |     | Max        | Average    | MLP-WS     | MLP-P      | Ours       |
> |-----|------------|------------|------------|------------|------------|
> | Acc | 69.2 (0.4) | 69.7 (0.3) | 70.7 (0.4) | 73.7 (0.5) | 77.2 (0.3) |
> | F1  | 29.2 (1.1) | 25.0 (0.5) | 32.8 (0.6) | 32.7 (1.0) | 34.0 (0.6) |
>
>
>
> > *Q9: Do you have any thoughts on why distilling features alone is better than distilling both features and logits?*
>
> Distilling both features and logits requires adapting the student network’s both backbone and classifier head as in MAML[21]. However, [78] have discovered that during fine-tuning, the representations of the network hardly change and thus the adapted model performs badly on domain shift tasks. Instead, in order to leverage representation change rather than representation reuse,  BOIL[45] forces the backbone to update while keeping the classifier head fixed. Also, BOIL empirically shows significant improvements over MAML, particularly on cross-domain tasks. For our task, since the class labels are unchanged across domains, we argued that adapting only the backbone should be as effective as BOIL. We empirically showed that BOIL is also more effective under our meta-knowledge distillation framework.
>
> > *Q10: There are quite a few sentences with incorrect grammar (in the Intro and throughout the paper), such as “Our contributions are manifold”. I recommend proofreading.*
>
> Thanks for the suggestion. We will change the sentence from “Our contributions are manifold” to “Our contributions are as follows”. Also, we will add extra rounds of proofreading and avoid incorrect grammar in the final version.
>
> > *Q11: In the Related Work section, ideally each paragraph should include direct contrast of prior works to the proposed method, to make the differences clear.*
>
> In fact, ARM is the only prior work in Test-time adaptation in DG, and we have included direct and comprehensive contrast between our method and ARM in L42 - 51 and L63 - 69. In the related work, we mainly introduce the problem settings and describe the background of elements used in our proposed method in each paragraph.
>
> > *Q12: moving Discussion on Social Impact to the main paper*
>
> Thanks for the suggestion, we will follow the guideline and include them in the main paper.
>
> **References:**
>
> [78] Raghu et al. "Rapid learning or feature reuse? towards understanding the effectiveness of MAML." ICLR, 2020
>
> [79] Yao et al. "Meta-learning with fewer tasks through task interpolation." ICLR, 2022.

---

> > ### Comment · Reviewer_cPHa · 2022-08-08
> > **Post-rebuttal comments**
> >
> > I thank the authors for the detailed responses, they have addressed most of my concerns. I expect the presented clarifications/changes to be incorporated in the paper, including the additional transformer-based baselines in Table 7.
> > I still think the Related Work section needs more careful structure w.r.t. providing clear contrast between prior work and the proposed work, due to the multitude of related yet slightly distinct problem statements. I hope the authors are willing to improve the RW for the benefit of the readers less familiar with the entire problem scope: e.g. add one sentence at the end of each paragraph positioning your work against the presented ones (what is similar, what is distinct).

---

> > > ### Author Response · Authors · 2022-08-09
> > > **Thank Reviewer cPHa, please see revision**
> > >
> > > Dear Reviewer cPHa,
> > >
> > > We thank you for the valuable time and comments to make our paper stronger. We have incorporated the contents in the rebuttal to the main paper and modified the Related Work. Please refer to the recent submitted revision and supplementary material. The following are the main changes in the paper:
> > >
> > > > *Related work*
> > >
> > > Please refer to Section 2 for detailed changes of Related Work. We modified the paragraphs and added some direct comparison between prior works and our proposed method.
> > >
> > > > *Q2: Concerns regarding the problem setting*
> > >
> > > Please refer to L40-43.
> > >
> > >
> > > > *Q3: How much data used for adatation process for each dataset*
> > >
> > > Please refer to L275-276.
> > >
> > >
> > > > *Q5: Detailed experimental setting*
> > >
> > > Please refer to Section G in the supplementary material.
> > >
> > >
> > > > *Q7: Clarification on increasing number of experts*
> > >
> > > Please refer to L334-338.
> > >
> > > > *Q8: Ablation in aggregators*
> > >
> > > Please refer to L355-360 and Secion C in the supplementary material.
> > >
> > > > *Q12: Move Social Impact to the main paper*
> > >
> > > Please refer to L390-396.
> > >
> > >
> > > Please let us know if any concern remains.
> > >
> > > Best,
> > >
> > > Authors of Paper 8220

---

> > > > ### Comment · Reviewer_cPHa · 2022-08-09
> > > > **Thank you!**
> > > >
> > > > The revisions look great to me.

---

> ### Author Response · Authors · 2022-08-02
> **Author Response to Reviewer cPHa (Part 1/2)**
>
> Thanks for taking the time to review our paper and provide insightful feedback! We address your comments and questions below:
>
> > *Q1: Since it is not stated clearly, is it correct that all the labels are shared across the source and target domains in this work?*
>
> Yes, the label space of both source and target domains are the same which is the default setting in Unsupervised Domain Adaptation (UDA) and Domain Generalization (DG). We have described such a setting with a formal definition of domain shift tasks in L126 - 128.
>
> > *Q2: The data examples from a target domain might be unavailable(L35) in Domain Generalization, but the author also mentions it can be operated in the setting where unlabeled target data is available. I was hoping the authors plan to address that or make the presentation clearer on that matter*
>
> In Domain Generalization, data samples from the target domain are unavailable during training but can be available at test time (as the target data has to be available for evaluation). Test-time adaptation with Domain Generalization,  as the problem setting we are aiming to solve, follows the limitation on the target domain during training in DG but allows unlabeled data from the target domain can be exploited to adapt the model at test time. A real-world application of such a setting is deploying a trained model to an unknown surveillance camera, where each camera can be treated as a domain. Then, before making the inference on images, the trained model can be adapted using the unlabeled images collected from the unknown camera. To make it clear, we will revise L40-42 as follows:
>
> "Test-time adaptation with DG [72] allows the model to exploit the unlabeled data during testing to overcome the limitation of using a flawed generic model for all unseen target domains. In ARM [72], meta-learning is utilized for training the model as an initialization such that it can be adapted using the unlabeled data from unseen target domain before making the final inference."
>
> > *Q3: How much target data is indeed used for adaptation by the proposed method? Would it have been good to include those details for all the datasets in the main paper?*
>
> Thanks for the suggestion. We will add “Specifically, we set the number of examples for adaptation at test time = {24, 64, 75, 64, 64} for iWildCam, Camelyon17, RxRx1, FMoW, and Poverty Map, respectively” in L281. We observed that, on the validation set in iWildCam, adapting 24 images for each target domain already yields good performance. For other datasets in WILDS, increasing the number of images may result in additional subtle improvement.
>
> > *Q4: How much target data is used for adaptation by competing baseline(s)? Is the comparison fair?*
>
> In Table 1, only ARM[72] adapts the model with unlabeled data from the target domain at test time. As described in L317-319, ARM needs to make an adaptation before inference on every example. As a result, the number of data used for test-time adaptation in ARM equals the total size of the entire dataset from the target domain. In contrast, our method only takes a small fixed-sized batch of data and thus requires much less computational cost for adaptation, as reported in Table 3.
>
>
> > *Q5. Detailed experimental setting. It would be good to clarify for the readers not closely familiar with WILDS if the author can detail more on the experimental settings. For example, what’s the number of sources and target domains respectively? How does the dataset split the validation set and the testing set? Is it the validation set labeled?*
>
> Thanks for the suggestion. To make readers understand the experimental setting more clearly, we will add the following contents regarding more details of official WILDS in Section F in the supplementary material.
>
> The official WILDS dataset contains training, validation, and testing domains which we use as source, validation target, and test target domains. The validation set in WILDS contains held-out domains with labeled data that are non-overlapping with training and testing domains. To be specific, we first use the training domains to pre-train expert models and meta-train the aggregator and the student prediction model and then use the validation set to tune the hyperparameters of meta-learning. At last, we evaluate our method with the test set. The official domain splits in WILDS datasets are as follows (train-source/val-target/test-target domains):
> * IWildCam:  243/32/48
> * Camelyon: 30/10/10
> * RxRx1: 33/4/14
> * FMoW:  55/15/15
> * Poverty: 26-28/8-10/8-10. Noted, the number of domains varies slightly from the fold to the fold for Poverty.

---

### Official Review · Reviewer_RtJB · 2022-07-11

**Rating:** 6
**Confidence:** 4
**Soundness:** 3 good
**Presentation:** 3 good
**Contribution:** 2 fair

**Summary:**

The proposed method addresses test-time adaptation as knowledge distillation from mixture of experts. Each expert is learned within each source domain as teachers. During testing time, a set of unlabeled sample is used to query the knowledge from the teachers. A transformer-based model is used to integrate knowledge from experts and meta-learning is used to encourage the aggregation for useful knowledge and student for good adaptation.

**Questions:**

See weaknesses.

What are the comparison results of the proposed method with recent SOTA on the datasets in [67]?

**Limitations:**

Yes.

**Strengths And Weaknesses:**

Strengths.

The idea is interesting. Distilling knowledge from different sources is not new, but the knowledge distillation and the meta learning seem novel to me. The paper is properly presented and some constrained real-word settings are also considered.

Weaknesses.

The major weakness is the lack of comparison with prior work on standard datasets. For example [67] (and more recent ones) and the datasets used by [67]. If the authors are willing to validate their method on these standard datasets, more recent works can be compared. Without comparing with those methods, it is hard to evaluate the performance of the method.


-------------------------- Post Rebuttal -----------------------------------------

My concerns regarding to comparison with [67] on standard datasets have been resolved. I raise my score to weak accept.

---

> ### Author Response · Authors · 2022-08-02
> **Author Response to Reviewer RtJB (Part 2/2)**
>
> > *Q2: Lack of comparison with prior work on standard datasets. E.g. the datasets used by [67]*
>
> ARM is the SOTA in TTA-DG. It was evaluated on 5 standard, realistic and large-scale domain shift datasets in the WILDS benchmark [33]. WILDS covers both classification and regression tasks using 5 evaluation metrics including accuracy, worst-case accuracy, Macro F1, Pearson correlation, and worst-case Pearson correlation. We strictly followed the experimental setting in ARM and would argue that our experimental results and comparison with ARM on the WILDS dataset are faithful and meaningful.
>
> WILDS is a more appropriate and challenging benchmark over the datasets in [67] due to: 1) the datasets in [67] are initially used for UDA, not TTA-DG. 2) only classification tasks and accuracy metrics are used in [67]. 3) the dataset in [67] is smaller scaled (the number of domains, categories), less close to real-world scenarios, and less imbalanced across domains. A detailed comparison of datasets in WILDS and datasets used in [67] is shown in Tab. 1.
>
> ***Table 1. Datasets in and [67] and WILDS***
> |Dataset|Year|Images|Categories|Domains|Imbalanced Ratio*|Descriptions|
> |-|-|-|-|-|-|-|
> |Digital-5|~|100k|10|5|~|digit
> |Office-Caltech10|2012|2.5k|10|4|~|office|
> |PACS|2017|9.9k|7|4|3.59|animal, stuff|
> |DomainNet|2019|569k|345|6|2.35|common objects|
> |iWildCam|2020|203k|182|323|3490|animal|
> |Camelyon17|2018|450k|2|50|38.57|tissue
> |RxRx1|2019|126k|1139|51|1.0|cell|
> |FMoW|2018|524k|62|16 * 5|2891|satellite|
> |PovertyMap|2020|19k|regression|23 * 2|15.05         |satellite|
>
> *Imbalanced Ratio = max number of samples/min number of samples across training domains
> *FMoW has 16 and 5 domains across time and geographical locations, respectively. Poverty has 23 and 2 domains across locations and rural/urban areas.
>
> However, to further evaluate our method, we follow [76] to compare our method with prior DG methods on the two most realistic datasets in [67]: DomainNet [75] and PACS [77]. Tab. 2 and Tab. 3 show the results. In DomainNet, our method performs the best on all experimental settings and outperforms recent SOTA significantly in terms of the average accuracy (+2.7). [79] has discovered that the lack of a large number of meta-training episodes leads to the meta-level overfitting/memorization problem. To our task, since PACS has 57x less number of images than DomainNet and 80x less number of domains than iWildCam, the capability of our meta-learning based method is hampered by the less diversity of episodes. As a result, we outperform other methods on 2 out of 4 experiments but still achieve the SOTA in terms of the average accuracy. We will add those experimental results to the supplementary material in the revision.
>
> ***Table 2: Evaluation on DomainNet***
> | |**clip**|**info**|**paint**|**quick**|**real**|**sketch**|**avg**|
> |-|-|-|-|-|-|-|-|
> | ERM|58.1 (0.3)|18.8 (0.3)|46.7 (0.3)|12.2 (0.4)|59.6 (0.1)|49.8 (0.4)|40.9|
> |IRM|48.5 (2.8)|15.0 (1.5)|38.3 (4.3)|10.9 (0.5)|48.2 (5.2)|42.3 (3.1)| 33.9|
> |Group DRO|47.2 (0.5)|17.5 (0.4)|33.8 (0.5)|9.3 (0.3)|51.6 (0.4)|40.1 (0.6)|33.3|
> |Mixup|55.7 (0.3)|18.5 (0.5)|44.3 (0.5)|12.5 (0.4)|55.8 (0.3)|48.2 (0.5)|39.2|
> |MLDG|59.1 (0.2)|19.1 (0.3)|45.8 (0.7)|13.4 (0.3)|59.6 (0.2)|50.2 (0.4)|41.2|
> |CORAL|59.2 (0.1)|19.7 (0.2)|46.6 (0.3)|13.4 (0.4)|59.8 (0.2)|50.1 (0.6)|41.5|
> |DANN|53.1 (0.2)|18.3 (0.1)|44.2 (0.7)|11.8 (0.1)|55.5 (0.4)|46.8 (0.6)|38.3|
> |MTL|57.9 (0.5)|18.5 (0.4)|46.0 (0.1)|12.5 (0.1)|59.5 (0.3)|49.2 (0.1)|40.6|
> |SegNet|57.7 (0.3)|19.0 (0.2)|45.3 (0.3)|12.7 (0.5)| 58.1 (0.5)|48.8 (0.2)|40.3|
> |ARM|49.7 (0.3)|16.3 (0.5)|40.9 (1.1)|9.4 (0.1)|53.4 (0.4)|43.5 (0.4)|35.5|
> |Ours|**63.5 (0.2)**|**21.4 (0.3)**|**51.3 (0.4)**| **14.3 (0.3)**|**62.3 (1.0)**|**52.4 (0.2)**|**44.2**|
>
>
> ***Table 3: Evaluation on PACS***
> | |**Art**|**Cartoon**|**Photo**|**Sketch**|**Avg**|
> |-|-|-|-|-|-|
> |ERM|84.7 (0.4)|80.8 (0.6)|97.2 (0.3)|79.3 (1.0)|85.5|
> |CORAL|**88.3 (0.2)**|80.0 (0.5)|**97.5 (0.3)**|78.8 (1.3)|86.2|
> |Group DRO|83.5 (0.9)|79.1 (0.6)|96.7 (0.3)|78.3 (2.0)|84.4|
> |IRM|84.8 (1.3)|76.4 (1.1)|96.7 (0.6)|76.1 (1.0)|83.5|
> |ARM|86.8 (0.6)|76.8 (0.5)|97.4 (0.3)|79.3 (1.2)|85.1|
> |Ours|86.1 (0.2)|**82.5 (0.5)**|96.7 (0.4)|**82.3 (1.4)**|**86.9**|
>
> **Reference:**
>
> [75] Peng et al. "Moment matching for multi-source domain adaptation." ICCV, 2019.
>
> [76] Gulrajani et al. "In Search of Lost Domain Generalization". ICLR, 2021
>
> [77] Li et al. "Deeper, broader and artier domain generalization." ICCV, 2017.
>
> [79] Yao et al. "Meta-learning with fewer tasks through task interpolation." ICLR, 2022.

---

> ### Author Response · Authors · 2022-08-02
> **Author Response to Reviewer RtJB (Part 1/2)**
>
> Thanks for providing insightful feedback! We address your concerns below:
>
> > *Q1: Missing the comparison with recent SOTA on the datasets in [67]? It is hard to evaluate the performance of the method.*
>
> There is a fundamental discrepancy in problem settings between [67] and ours, so it is unfair to compare with the methods listed in [67]. Specifically, [67] tackles the problem of unsupervised domain adaptation (UDA), while we are aiming for test-time adaptation (TTA) with domain generalization(DG). In the later setting, ARM [72] is the SOTA that we have compared.
>
> In fact, we have explained the differences among UDA(L29-35, L86-87), DG(L36-39, L93-95), and TTA-DG(L40-42, L99-100) in both INTRODUCTION and RELATED WORK. In summary:  UDA assumes that labeled data from the source domain(s) and unlabeled data from the target domain are both present during training. In contrast, DG is a more realistic and challenging setting where only data from the source domain(s) are available but any prior of the target domain is unknown during training. TTA-DG strictly follows the data constraints in DG but allows the unlabeled data from the target domain to be exploited to adapt the model at test time. Therefore, it is unfair to directly compare methods of UDA in [67] with our proposed method in TTA-DG.

---

> ### Author Response · Authors · 2022-08-07
> **Further discussion with Reviewer RtJB**
>
> Dear Reviewer RtJB,
>
> We would like to thank you for your time reviewing our paper and for providing meaningful comments. According to the initial review, we received borderline reject due to the missing evaluation on benchmarks used in [67]. In the rebuttal, we evaluate and compare our method with other DG methods on DomainNet and PACS datasets. We have also shown the superiority of our method. We hope that our responses can address your concerns. If not, we would sincerely appreciate the chance to discuss this further with you. Please let us know if you have any further concerns.
>
> Best,
>
> Authors of Paper 8220

---

> > ### Comment · Reviewer_RtJB · 2022-08-07
> > **Concerns Resolved**
> >
> > Thanks for the authors' responses and clarification. My concerns are well resolved.

---

> > > ### Author Response · Authors · 2022-08-07
> > > **Thank Reviewer RtJB**
> > >
> > > Dear Reviewer RtJB,
> > >
> > > We would like to thank you for taking the time to read our responses. We are truly happy that our rebuttal resolves your concerns. We would like to kindly ask Reviewer RtJB to consider updating the scores if applicable.
> > >
> > > Best,
> > >
> > > Authors of Paper 8220

---

### Author Response · Authors · 2022-08-02
**Thanks to all the reviewers!**

We thank reviewers for their positive feedback on our paper which contributes to:
* Meta-DMoE, a novel unsupervised test-time adaptation framework in Domain Generalization, which enables the model to fast adapt to unseen target domains via distilling the knowledge from multiple domain-specific models
* state-of-the-art results on the standard large-scale distribution shift benchmark WILDS

All the reviewers agreed that our work is an effective solution to alleviate the problem of domain shift. They commended the novelty of the approach [RtJB, cPHa, RHgP], the extensiveness of the experiments [cPHa, RHgP], and the additional studies concerning the constrained computational cost and data privacy [RtJB, cPHa, RHgP].

Primarily, reviewers requested additional experiments and clarifications on existing experiments. Thus, in the rebuttal, we have detailed the experimental settings[cPHa], and also have performed new experiments [RtJB] that demonstrate the effectiveness of our proposed method on datasets in [67].

We believe that these experiments and our responses address all the questions, issues, and comments.

**References:**

[67] Luyu Yang, Yogesh Balaji, Ser-Nam Lim, and Abhinav Shrivastava.  Curriculum manager for source selection in multi-source domain adaptation. In European Conference on Computer Vision, 2020.

---

### Meta-Review · Area_Chair_Z19T · 2022-08-26

**Recommendation:** Accept
**Confidence:** Certain

**Metareview:**

This paper addresses the DG-TTA problem setting, drawing inspiration from the recent Adaptive Risk Minimisation (ARM). It goes beyond ARM to introduce a mixture of experts and distills from those mixture of experts during adaptation. Reviewers agree the MoE idea makes sense, and the distillation-based adaptation is interesting and novel enough, and they appreciated the good results and evaluation. Concerns included various writing clarity issues and evaluation on other datasets, but these questions were generally resolved in the rebuttal. Given that concerns were resolved and all reviewers were positive, I recommend accept.


**Award:**

No

---

### Decision · Program_Chairs · 2022-09-14

Accept